Evidence of an upper entrainment limit for walking with fractal auditory stimuli

Power Cecilia R. crpower@yorku.ca 1
Sorensen Kristen L. 2
Drake Janessa D.M. 1
Gage William H. 1
1 School of Kinesiology and Health Science, York University , Toronto , Ontario , Canada
2 School of Allied Health Professions, Keele University , Keele , Staffordshire , United Kingdom
Vieira Marcus
Electronic publication date: 2025 Oct 27
Publication date: 2025
Volume: 13
Electronic Location ID: e20176
Received 2025 Mar 27; Accepted 2025 Sep 12
Copyright: ©2025 Power et al.
Copyright year: 2025
Copyright holder: Power et al.
License: This is an open access article distributed under the terms of the Creative Commons Attribution License, which permits unrestricted use, distribution, reproduction and adaptation in any medium and for any purpose provided that it is properly attributed. For attribution, the original author(s), title, publication source (PeerJ) and either DOI or URL of the article must be cited.
License URL: https://creativecommons.org/licenses/by/4.0/

Keywords: Gait analysis, Gait variability, Detrended fluctuation analysis, Fractal scaling index, Nonlinear analysis, Entrainment, Auditory metronome, Gait adaptability, Fractal entrainment threshold

Funding: The authors received no funding for this work.

==============================
Background

Variability exists in all biological signals, and in human gait research it has been found to be an indicator of neuromuscular system functioning. Detrended fluctuation analysis (DFA), a nonlinear method used to quantify the strength of long-range correlations in the temporal structure of stride-to-stride gait variability, has revealed gait differences in certain populations that are not observed with traditional linear measures like standard deviation. Previous research suggests that humans can adapt gait patterns to match different variability structures through sensory cues, such as auditory metronomes. However, the upper limits of adaptability and the strength of long-term correlations in gait variability remain unclear. Exploring these limits not only deepens our understanding of neuromuscular control mechanisms but could also inform the design of targeted interventions, such as rehabilitation strategies, to enhance motor control in clinical populations. The aim of this study was to investigate the possible upper limits of long-term correlations in stride-to-stride gait variability, characterized by the fractal scaling index (FSI) using DFA.

Methods

Fourteen healthy young adults (age 25 ± 3 years; seven females) completed seven treadmill walking trials at a fixed, self-selected speed. The first trial was uncued, and during the remaining six trials participants timed their steps to an auditory metronome with FSI ranging between 1.00 and 1.25. Gait FSI, velocity, stride time, cadence, and the time difference between heel contact and the associated metronome “tones” were calculated.

Results

Uncued gait FSI averaged 0.76 (±0.1). As the metronome FSI increased from 1.00 to 1.15, gait FSI approximated 1.00. Beyond 1.15 (metronome FSI values of 1.20 and 1.25), gait FSI dropped below 0.70, resembling uncued walking. Other gait measures remained unchanged. These findings suggest an upper gait FSI limit of approximately 1.00 during entrainment to metronome FSI values <1.20, beyond which adaptability diminishes.

Conclusions

This study establishes the upper entrainment limit for gait FSI during synchronization with fractal auditory stimuli, with implications for designing effective gait rehabilitation interventions targeting specific variability patterns.

Introduction

Gait analysis is a widely used tool to evaluate functional performance, including applications in pre- and post-surgery assessments, disease progression monitoring, and sports performance. Traditionally, gait analysis has focused on peak-dependent metrics such as mean values and associated standard deviations for assessing parameters like walking speed, stride length, and swing time, alongside numerous other kinematic and kinetic measures. While these linear metrics provide value, literature suggests that the “noise” within biological signals offers significant insight into the systems that produce them (Hausdorff, 2005; Stergiou, 2016; Stergiou & Decker, 2011). Specifically, variability in gait, stemming from non-linear and deterministic origins, is an important indicator of neuromuscular control and adaptability (Stergiou, 2016). Over the past few decades, a new category of gait-related measurement approaches has emerged, promising additional insights into gait control and the effects of disease, injury, and rehabilitation. Unlike traditional metrics, these newer approaches, including detrended fluctuation analysis (DFA), characterize the nonlinear, dynamic aspects of gait and have demonstrated potential as sensitive indicators of change or variability between groups, with promising clinical relevance (Di Bacco & Gage, 2024; Hausdorff et al., 1995; Hausdorff, 2007; Ravi et al., 2020). However, researchers and clinicians are still in the early stages of understanding these measures. Questions remain about what these metrics reveal regarding functional performance, how they reflect changes over time, and how they relate to the neural and biomechanical constructs underlying gait behavior.

This paper focuses on a particular non-linear measure, the fractal scaling index (FSI), a measure which quantifies how similar a time series is across different time scales (Buzzi et al., 2003; Di Bacco & Gage, 2024; Hausdorff et al., 1995; Kiriella et al., 2020; Rhea et al., 2014; Terrier & Dériaz, 2011). By examining stride time variability, FSI helps us understand the neuromechanical control of gait and stability. It reflects long-range correlations in stride-to-stride gait variability, particularly in the inter-stride intervals (ISI) or the times between consecutive strides (Buzzi et al., 2003; Hausdorff, 2007; Stergiou & Decker, 2011). In healthy gait, ISI does not change randomly. Rather, changes in ISI exhibit a high degree of structure, with some persistence in the direction of fluctuations in step timing. The relationship between consecutive steps diminishes gradually over time in a “fractal-like” or “scale-free” pattern (Hausdorff, 2005; Hausdorff et al., 1995; Kiriella et al., 2020). To put it simply, this structured variability is not chaotic, but instead, it reflects a highly adaptable and resilient system that allows us to adjust to changing environments, such as navigating obstacles or adjusting pace. In a way, the variability in walking reflects a purposeful flexibility that is essential for healthy function. A lack of such variability may lead to difficulties with balance or navigating cluttered environments, whereas excessive variability might cause instability and fall risk. FSI values describe how the correlations decay over time and are characterized as different types of noise: 0.50 is white noise (WN; a type of randomness), 1.00 is pink noise (PN), and 1.50 is red noise (RN) (Hausdorff et al., 1995; Kiriella et al., 2020). Human gait typically demonstrates gait FSI values between 0.60 and 1.00 (Hausdorff, 2007; Hausdorff et al., 1995; Kiriella et al., 2020). Lower values, closer to white noise, are observed in groups with a higher risk of falling, like older adults and people with Huntington’s or Parkinson’s Disease (Goldberger et al., 2002; Harrison & Earhart, 2023; Hausdorff et al., 1997; Kiriella et al., 2020). This insight into the structure of gait variability offers potential not only for understanding functional health but also for diagnosing gait-related issues in clinical populations.

To explore the nature and potential of FSI, researchers have examined whether fractal sensory stimuli can influence gait patterns (Frame et al., 2024; Hove et al., 2012; Hunt, McGrath & Stergiou, 2014; Kiriella et al., 2020; Marmelat et al., 2014; Rhea et al., 2014; Uchitomi et al., 2013). For example, metronomes designed to emit fractal-like tones—mimicking the variability of healthy gait—offer a unique tool to study gait adaptability. These auditory cues encourage the adjustment of stride timing to match the fractal pattern of the stimuli, providing insight into the mechanisms underlying flexible and adaptive walking (Roerdink et al., 2015; Piergiovanni & Terrier, 2023; Thaut, McIntosh & Hoemberg, 2015). By understanding how healthy individuals respond to such cues, we can identify strategies to assist clinical populations whose walking patterns are disrupted by illness or injury (Frame et al., 2024; Thaut, McIntosh & Hoemberg, 2015). This line of research also holds promise for improving interventions aimed at enhancing balance, stability, and recovery in diverse environments.

Although studies suggest that fractal auditory patterns can change walking patterns, there appear to be limits to this adjustment. For example, even when using metronome patterns with high FSI values (like RN with an FSI of 1.3), participants have not consistently reached an FSI of 1.00 or higher (Hunt, McGrath & Stergiou, 2014; Kiriella et al., 2020). These results suggest there might be a limit before 1.30 to how much walking patterns can change to match these fractal stimuli (Hunt, McGrath & Stergiou, 2014; Kiriella et al., 2020). Thus, this paper focuses on the extent to which the value of the FSI can be experimentally influenced among young, healthy participants, with a specific focus on the upper limits of FSI. Given that previous studies have not fully explored the impact of metronomes with FSI values between 1.00 and 1.30, this work seeks to bridge that gap. By examining whether gait FSI values can reach or exceed 1.00 in response to stimuli with FSI values lower than 1.30, we aim to better understand the adaptability of the human gait system and bridge the gap between 1.00 and 1.30 that previous studies did not examine. The hypotheses of the study were (1) that gait FSI values will equal values of 1.0 or greater in response to stimuli which have FSI values close to 1.0, and that (2) gait FSI values will significantly differ across stimulus conditions, with an upper limit beyond which individuals are no longer able to entrain their gait to the auditory stimuli. In addition to examining changes in gait FSI, this study also quantified entrainment error (the absolute difference between stimulus and gait FSI), timing error (the time between tone onset and heel contact), and basic spatiotemporal gait measures such as cadence, stride time, and velocity to explore how nonlinear and linear metrics respond across stimulus conditions. This work addresses a critical gap in our understanding of the very nature of the FSI, the use of which seeks to reveal elements of the control of gait behaviour and is essential to establishing the validity of the measure in terms of future clinical utility.

Materials & Methods

Participants

Fourteen young healthy adults (aged 25.3 ± 2.7 years; seven females) volunteered to participate. Sample size was determined using G*Power for a repeated-measures ANOVA (within-subjects, seven levels), assuming α = 0.05, and a desired power level of 0.90. A large effect size (f = 0.4; Cohen, 1988) was assumed, based on the results of Kiriella et al. (2020). The analysis suggested that n = 9 would be sufficient and therefore that 14 participants would be more than sufficient to detect a within-subject effect. The study protocol was approved by the York University Human Participants Review Committee (HPRC) (approval number STU2018-123), and all participants provided written informed consent. Exclusion criteria included any history of neurological disorders or injuries; musculoskeletal disorders, injuries, or pain within the past 12 months; any auditory impairments; and an inability to walk unassisted for periods of 20 min or more.

Protocol

Participants completed a single laboratory session, during which gait data, as well as demographic information including age, sex, height, weight, and ankle width, were recorded. Each participant was equipped with infrared reflective markers (Fig. 1) for motion capture, which was recorded using seven VICON motion capture cameras (MX40; VICON, Denver, CO, USA) sampling at 100 Hz. Safety devices, including a treadmill shutoff cord and a harness suspended from the ceiling, were used to ensure participant safety throughout the session.

Figure 1 Marker name and corresponding anatomic location of each marker for use with 3D motion capture.

The session began with a preferred walking speed (PWS) test, followed by an uncued baseline walking trial, and six cued walking trials (Fig. 2). The PWS test was conducted according to methods used in previous studies (Kiriella et al., 2020; Marmelat et al., 2014). Briefly, the average PWS of both an increasing (PWSi) and a decreasing (PWSd) PWS were obtained by asking participants to indicate when they were walking at a comfortable pace while blinded to the treadmill belt speed and the researcher either increased the belt speed (PWSi) or decreased the belt speed (PWSd) by 0.1 mph every 10 s. Average preferred walking speed across participants was 2.3 (+- 0.4) mph. Next, participants completed an uncued baseline walking trial, during which they walked at their PWS for at least 256 strides while motion capture data were collected (Hausdorff, 2005; Kiriella et al., 2020; Marmelat et al., 2014). Following the baseline trial, participants engaged in six cued walking trials, each involving auditory fractal stimuli.

Figure 2 Procedural sequence flow chart.

Sequence occurs numerically beginning with block 1 and ending with block 5. Rest periods were given between each block, between each walking trial, and whenever otherwise needed by the participant. The asterisk (*) in blocks 3 and 5 indicate motion capture collection. Stimuli (represented by their FSI values) used individually for each trial within block 5 were performed in random order.

Six fractal auditory stimuli were generated using a MATLAB® algorithm (R2018b; The MathWorks, Natick, MA, USA), based on methods described in previous studies (Kasdin, 1995; Kiriella et al., 2020; Marmelat et al., 2014). This algorithm used the participant’s baseline mean and standard deviation of inter-stride intervals (ISI) to create a white noise (WN) signal. The signal was then transformed into the frequency domain, where its power spectrum components were multiplied by either 1/√f or 1/f to create pink noise (PN) or red noise (RN) signals, respectively. Afterward, the signal was transformed back to the time domain. Each stimulus consisted of 256 data points, or ”tones”, each lasting 10 msec, with fluctuating inter-tone time intervals designed to yield auditory stimuli with FSI values of approximately 1.00, 1.05, 1.10, 1.15, 1.20, and 1.25.

Prior to motion capture data collection, the FSI values of each auditory stimulus were verified using DFA, as described in previous studies (Bruijn et al., 2013; Chau, 2001; Damouras et al., 2010; Peng et al., 1995). Briefly, the ISI time series of length N was integrated and divided into equal-length intervals, and the fluctuation around the line of best fit was calculated using the root mean square (RMS) for each interval (n). This process was repeated for varying interval lengths, and FSI was calculated as the slope of the log–log plot of average RMS values versus log n values.

Before each cued walking trial, a sample auditory metronome was played via speakers, and participants were asked if they could clearly hear the tones. Participants were then instructed to synchronize their right heel contact (HC) with the beat of the auditory stimulus. They were given 1 min for practice. Each trial lasted approximately 5 min, or the time for 256 strides (Kiriella et al., 2020; Marmelat et al., 2014). The auditory stimuli were presented in a randomized order and 1- to 2- minute rest breaks were provided between trials to minimize the potential effect of fatigue. The stimulus tones were recorded synchronously with the motion capture data, with the speakers connected to the Vicon system via a custom-made jack-to-BNC cable.

Fractal scaling index analysis

Gait FSI was determined from ISI data processed in Visual3D (v5, C-motion, USA) using unfiltered anterior-posterior (A/P) velocity derived from horizontal displacement data of the marker placed on the right calcaneus (RCAL). Instances of HC were determined by using the “Event_Threshold” pipeline to find frames just prior to descending zero crossings of the velocity signal in the axis of progression, similar to what was done by Zeni, Richards & Higginson (2008). For each trial, times between consecutive HC events were calculated to construct ISI time series, and gait FSI values were calculated in MATLAB® using the ISI time series and DFA method described earlier (Chau, 2001; Damouras et al., 2010; Kiriella et al., 2020; Peng et al., 1995). Also, FSI entrainment error was calculated for each trial by determining the absolute difference between stimulus FSI and gait FSI values.

Time difference between heel contact and stimulus onset analysis

As analog stimulus data and motion capture data were collected simultaneously at different sampling rates (1,000 Hz and 100 Hz respectively), motion capture velocity data were upsampled to 1,000 Hz using the “interp()” function in MATLAB® to linearly interpolate between original consecutive data points. Stimulus data were upsampled to ensure that critical time points for the time series analyses were not lost. Using upsampled velocity data, times of HC were determined in MATLAB® using the zero crossings method described above. Also, using stimulus data, times of stimulus tone onset were determined in MATLAB® using the “findpeaks()” function to determine times of peak values whenever the signal crossed a specified 0.01V threshold value (Kiriella et al., 2020). Times of HC were subtracted from respective tone onset times to determine the time difference between HC and stimulus onset (TimeTone-HC). TimeTone-HC means were calculated using absolute values.

Kinematics analysis

Average stride time was the average of right calcaneus (RCAL) ISI data constructed earlier (described above). Displacement data from the RCAL marker were filtered using a dual-pass, low-pass Butterworth filter with a cut-off frequency of 12 Hz (Stergiou, 2016). The zero crossings method was used again to determine right HC time data for RCAL filtered displacement data. Cadence was found by dividing the number of steps taken between the first and last HC by the time in minutes. Average walking velocity was also determined for each trial using filtered RCAL marker A/P displacement data in Stata (StataMP 16; StataCorp, College Station, TX, USA) following previous recommendations (Souza et al., 2017). Briefly, the first and second derivatives of displacement data were taken to respectively create A/P velocity and A/P acceleration data. Next, parts of the velocity curve that were either in a positive direction (i.e., foot swinging anteriorly) or the corresponding acceleration is above a threshold of 0.02 m/s2 to isolate the portion of the signal when the foot is on the belt of the treadmill, were removed. Using this treadmill velocity curve, we calculated the demeaned RMS and then added the RMS to the mean of the treadmill velocity. Lastly, the sum of the absolute value of the RMS and the mean treadmill velocity was used to represent the mean walking velocity.

Statistical analysis

Statistical analyses used IBM® SPSS statistics software (Version 25; IBM corporation, Armonk, NY, USA) with an alpha of 0.05 used to indicate significance. All dependent variables (gait FSI, entrainment error, TimeTone-HC, cadence, walking velocity, and stride time) were tested using a Shapiro–Wilk test of normality. Repeated measures analysis of variance (ANOVA) was used to compare the means for each dependent measure across the seven stimulus FSI conditions (baseline plus six metronome trials). In the case of statistically significant differences, Tukey post-hoc tests were used to explore differences across stimulus FSI levels. For gait FSI, box-and-whisker plots were constructed based on the FSI values calculated for each trial, with outliers defined as values lying beyond 1.5 times the interquartile range from the first or third quartile. Two participants were removed from the FSI analysis due to being outliers.

Results

Fractal scaling index

Gait FSI differed significantly across stimulus FSI levels (F(6,11) = 16.799, p < 0.001). Mean and standard deviation values for each stimulus FSI value (i.e., per trial) are presented in Table 1 and all gait FSI findings in Fig. 3. Tukey post-hoc analysis results are presented in Table 2 for the factor of stimuli FSI. The gait FSI values in uncued baseline trials were lower than gait FSI values in trials using stimuli with FSI values of 1.00, 1.05, or 1.15. Also, the gait FSI values in trials with stimuli FSI values of 1.00, 1.05, 1.10, and 1.15 were higher than gait FSI values in trials with stimuli FSI values of 1.20, and 1.25. Entrainment errors also differed significantly across stimulus FSI levels, with a trend whereby error increases as stimulus FSI increases. Mean and standard deviation entrainment error values per trial are presented in Table 1 and Tukey post-hoc analysis results are in Table 2. It was found that entrainment errors for stimuli FSI values of 1.00, 1.05, 1.10, and 1.15 were significantly lower than stimuli FSI values of 1.20 and 1.25 (F(5,11) = 7.625, p < 0.001).

Table 1 Average and standard deviation of dependent measure values for all subjects within each trial as indicated by respective stimulus fractal scaling index (FSI) values.

Stimulus FSI	Dependent measures	
	Gait FSI	Entrainment error	Cadence (steps/min)	Walking velocity (m/s)	Stride time (s)	
None	0.76
(±0.10)	–	100.92
(±9.77)	1.03
(±0.18)	1.20
(±0.11)	
1.00	0.99
(±0.12)	0.11
(±0.05)	100.17
(±9.44)	1.03
(±0.18)	1.21
(±0.11)	
1.05	1.01
(±0.10)	0.10
(±0.06)	100.26
(±9.46)	1.03
(±0.18)	1.21
(±0.11)	
1.10	0.91
(±0.22)	0.22
(±0.21)	100.60
(±9.50)	1.03
(±0.18)	1.20
(±0.11)	
1.15	1.03
(±0.17)	0.19
(±0.20)	100.15
(±9.66)	1.03
(±0.18)	1.21
(±0.11)	
1.20	0.68
(±0.12)	0.46
(±0.19)	100.15
(±9.32)	1.03
(±0.19)	1.21
(±0.11)	
1.25	0.67
(±0.11)	0.57
(±0.11)	100.47
(±9.49)	1.03
(±0.18)	1.20
(±0.11)	

Figure 3 Gait fractal scaling index (FSI) in response to varying stimulus FSI values.

A total of 4 outliers were excluded based on box and whisker plots constructed per each trial. Baseline, uncued gait FSI measurements are represented by dashed lines. Standard deviation bars are included for cued gait FSI. Star (⋆) indicates statistically significant differences.

Table 2 Post-hoc Tukey analysis p-values for the factor of stimulus fractal scaling index (FSI) on the dependent measures of gait FSI and entrainment error.

Stimuli FSI groups	Dependent measures	
	Gait FSI	Entrainment error	
None vs. 1.00	0.001*	–	
None vs. 1.05	<0.001*	–	
None vs. 1.10	0.182	–	
None vs. 1.15	<0.001*	–	
None vs. 1.20	0.898	–	
None vs. 1.25	0.775	–	
1.00 vs. 1.05	0.979	0.986	
1.00 vs. 1.10	0.594	0.990	
1.00 vs. 1.15	0.911	1.000	
1.00 vs. 1.20	<0.001*	0.042*	
1.00 vs. 1.25	<0.001*	0.004*	
1.05 vs. 1.10	0.146	1.000	
1.05 vs. 1.15	1.000	0.988	
1.05 vs. 1.20	<0.001*	0.006*	
1.05 vs. 1.25	<0.001*	<0.001*	
1.10 vs. 1.15	0.078	0.992	
1.10 vs. 1.20	0.009*	0.007*	
1.10 vs. 1.25	0.003*	0.001*	
1.15 vs. 1.20	<0.001*	0.039*	
1.15 vs. 1.25	<0.001*	0.004*	
1.20 vs. 1.25	1.000	0.969	
Notes.

P-values are listed by stimuli FSI value groupings whereby uncued trials are represented by “none”. Statistically significant differences (p < 0.05) are indicated by an asterisk (*).

Time difference between heel contact and stimuli onset

Due to incomplete stimuli data, two participants were excluded from analysis of the mean TimeTone-HC. We found no significant effect of stimulus FSI (F(5,11) = 1.272, p = 0.288). On average, TimeTone-HC wwas 0.103s (±0.057s), and participants showed positive differences (i.e., HC occurred before stimulus tone onset) 84.9% of the time.

Kinematics

All kinematic measure means and standard deviations across stimulus levels are presented in Table 1. No differences across different stimulus FSI levels were found for cadence (F(6,13) = 0.16, p = 1.000), walking velocity (F(6,13) < 0.001, p = 1.000), or stride time (F(6,13) = 0.016, p = 1.000).

Discussion

The results of this study provide important new insights into the upper limit of auditory stimulus FSI, specifically as it relates to long-range correlations in stride time series (i.e., the sequence of times between consecutive strides) to which healthy individuals can entrain. These findings have critical implications for designing and implementing therapeutic interventions that use fractal metronomes in gait training. For clinicians, it is now advisable to use metronomes with FSI values below a specific FSI threshold of 1.20, as our results suggest that entrainment may be compromised above this limit.

This study advances the understanding of how variability in gait has a structured, deterministic origin, as shown by nonlinear analysis techniques like DFA (Goldberger et al., 2002; Hausdorff, 2005; Kiriella et al., 2020; Ravi et al., 2020). Where natural gait variability in healthy populations demonstrates 1/f scaling (PN), there is a shift toward more random patterns (WN) with age or disease (Goldberger et al., 2002; Harrison & Earhart, 2023; Hausdorff et al., 1995; Hove et al., 2012; Kiriella et al., 2020). This finding is consistent with the baseline measures from our study, where an average FSI value of 0.76 (±0.10) was found for healthy participants.

As shown previously, fractal auditory stimuli can entrain gait FSI to match that of the stimulus (Hove et al., 2012; Hunt, McGrath & Stergiou, 2014; Kiriella et al., 2020; Rhea et al., 2014; Piergiovanni & Terrier, 2023). However, entrainment typically occurs only when the stimuli have either WN or PN structure, and gait FSI values rarely exceed 1.00 (Hunt, McGrath & Stergiou, 2014; Kiriella et al., 2020). In contrast, the current study demonstrates that individuals can achieve gait FSI values greater than 1.00 (see Table 1 and Fig. 3). This finding is significant because earlier studies did not explore FSI values between 1.00 and 1.30, a range used in the present study, which may explain why the upper limit we identified had not been previously revealed (Kiriella et al., 2020). Previous studies used RN stimuli with FSI values around 1.30, whereas our study found that on average gait FSI values of 1.00 or higher were reached during trials with stimuli FSI values of 1.05 or 1.15. Moreover, our study provided evidence of an entrainment limit at a stimulus FSI of 1.20, beyond which entrainment is no longer observed. This suggests that entrainment may be constrained by the range of 1/f scaling, which lies between 0.70 and 1.20 (Stergiou, 2016), indicating that individuals may not be able to entrain beyond this range. Importantly, identifying the limits of entrainment may also offer insights into how adaptability is embedded within gait patterns. While the ability to entrain to fractal stimuli provides a framework for designing effective therapeutic interventions, it also reflects the intrinsic functional adaptability of the locomotor system (Thaut, McIntosh & Hoemberg, 2015). The observed entrainment range could represent a physiological window in which adaptive responses to external perturbations are optimized, and deviations from this range might suggest impaired adaptability. For instance, individuals with movement disorders or impaired gait variability might show altered entrainment capabilities, potentially revealing underlying mechanisms of dysfunction. Conversely, therapeutic interventions designed within the entrainment range may enhance functional adaptation by leveraging the fractal dynamics of gait. This link underscores the potential for entrainment studies to advance our understanding of both the mechanisms of impairment and the parameters most conducive to rehabilitation.

Interestingly, as shown in Fig. 3, gait FSI values were greater in trials where the stimulus FSI was between 1.00 and 1.15 compared to values of 1.20 or 1.25. Additionally, as seen in Table 1 and Fig. 3, most standard deviations of gait FSI were between 0.10 and 0.15, but trials with stimuli FSI values of 1.10 and 1.15 exhibited larger standard deviations. This suggests that there were greater inter-individual differences in entrainment, with some individuals continuing to entrain at the 1.20 stimulus FSI, while others experienced decreased entrainment after 1.10. This, coupled with the observed decline in entrainment after the 1.20 stimulus FSI, indicates a potential adaptability limit for entrainment at the FSI value of 1.20.

In contrast to non-linear measures such as FSI, linear measures provide limited insight into the neuromuscular system’s nuanced responses to dynamic changes (Stergiou, 2016). Our study supports this distinction, as no significant differences were found in cadence, ISI, or walking velocity across the different stimulus FSI values, reinforcing the sensitivity of FSI to capturing complex gait variability. Linear measures remain valuable as essential baselines for standard clinical and biomechanical evaluations. The findings of this study also align with previous literature suggesting a lack of correlation between linear and nonlinear measures of gait (Bruijn et al., 2013; Buzzi et al., 2003; Damouras et al., 2010; Di Bacco & Gage, 2024; Hausdorff et al., 1995; Stergiou & Decker, 2011), confirming that changes in nonlinear measures may not be strongly associated with changes in linear measures across conditions with differing fractal stimuli. However, non-linear approaches, such as those leveraging fractal dynamics, offer deeper insights into gait’s adaptive and dynamic nature, especially under challenging conditions or therapeutic interventions. Advancing gait analysis requires evidence to standardize non-linear methods and demonstrate their utility across diverse populations. Interdisciplinary collaboration among clinicians, data scientists, and engineers is vital to developing accessible, interpretable, and clinically relevant tools. Future gait analyses could benefit from hybrid models that combine the strengths of linear and non-linear methods, enabling a more holistic understanding of gait for personalized diagnostics and interventions. Embracing this paradigm will make assessments smarter and more responsive to human movement’s complexities. Further research should explore the relationship between these measures and functional outcomes meaningful to patients, helping to determine which (or both) of these measures best reflect clinically relevant improvements in gait, ultimately guiding the development of more effective rehabilitation interventions.

The study does have several limitations that should be considered. These include the use of right foot cueing for all participants, absence of beat perception testing, lack of consideration for participants’ physical activity history, the time series length of 256 strides, and the use of a fixed-speed treadmill. Right-foot cueing was used to ensure consistency across participants; however, future research may consider exploring the influence of limb dominance on gait entrainment. While entrainment error and TimeTone-HC were measured, participants were not pre-screened for their ability to perceive beats. Additionally, prior experiences, such as musical training or participation in activities involving auditory movement cueing (e.g., dance or figure skating), may have influenced results and participants’ ability to entrain their gait to fractal stimuli. While longer trial lengths may yield more robust calculations and deeper insights, a trial length of 256 strides was selected to minimize potential fatigue and attention loss, ensure feasibility for clinical populations, and align with previous studies. Lastly, while a fixed-speed treadmill was used to minimize changes in gait FSI from speed variation, future studies could explore the effects of adaptive treadmills or over-ground walking on entrainment. Addressing these limitations in future research could provide deeper insights into gait entrainment to fractal stimuli.

Conclusions

This study showed that individuals can achieve optimal spatio-temporal patterns within their gait and entrain gait to have FSI values of 1.00. This is significant because previous studies only shown entrainment to values less than 1.00. Additionally, the findings of this study corroborate previous research that suggests an entrainment limit. This entrainment limit, or a limit for the neuromuscular system’s adaptability, was shown to occur at stimuli FSI values of approximately 1.20, after which gait FSI values approach uncued baseline levels. In our analysis, we did not find significant differences in linear measurements across trials. This suggests that changes in nonlinear measures may not be strongly associated with changes in linear measures across conditions with differing fractal stimuli. Overall, the findings of this study have provided new insights for understanding the neuromuscular control of gait. We have shown that an upper limit for matching gait to fractal patterns does exist, which allows scientists to better understand the extent to which walking patterns can be altered to fit an environment, and further can be used in developing interventions and rehabilitation treatments for gait. These results have provided insight into the extent to which the structure of variability, and thus the adaptability, of the system can be altered. These results underscore the importance of using both types of analyses to gain a comprehensive understanding of gait dynamics, as nonlinear measures appear to provide novel information to understanding gait behaviour.

Supplemental Information

Supplemental Information 1 Raw Data Values

Column 1 identifies the participant (with code names removed for full anonymity). Column 2 indicates the participant’s sex, where “1” represents female and “2” represents male. Column 3 specifies the individual’s preferred walking speed on the treadmill. Column 4 designates the trial condition, where “0” represents the baseline, “1” corresponds to the metronome with FSI 1.00, “2” corresponds to FSI 1.05, “3” corresponds to FSI 1.10, “4” corresponds to FSI 1.15, “5” corresponds to FSI 1.20, and “6” corresponds to FSI 1.25. The remaining columns contain the mean values of the variables listed in row 1 for each participant and trial.

Additional Information and Declarations

Competing Interests

Author Contributions

Human Ethics

Data Availability

The authors declare there are no competing interests.

Cecilia R. Power conceived and designed the experiments, performed the experiments, analyzed the data, prepared figures and/or tables, authored or reviewed drafts of the article, and approved the final draft.

Kristen L. Sorensen conceived and designed the experiments, authored or reviewed drafts of the article, and approved the final draft.

Janessa D.M. Drake conceived and designed the experiments, authored or reviewed drafts of the article, and approved the final draft.

William H. Gage conceived and designed the experiments, prepared figures and/or tables, authored or reviewed drafts of the article, and approved the final draft.

The following information was supplied relating to ethical approvals (i.e., approving body and any reference numbers):

The York University Human Participants Review Committee (HPRC) granted Ethical approval to carry out the study within its facilities (Ethical Application Ref: STU 2018-123).

The following information was supplied regarding data availability:

The data is available in the Supplemental File.

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
