# Peer review of "Evidence of an upper entrainment limit for walking with fractal auditory stimuli"

_PeerJ, doi:10.7717/peerj.20176_

## Round 0.1 · original submission · Major Revisions

The authors should consider all reviewers' comments. Additionaly, please explain the sample size used in the study.

Reviewer 1 ·

Basic reporting

There are a few areas that could benefit from further improvement.

Abstract,

Line 20 - Auditory cues make more sense; please consider replacing them.

Introduction,

A sufficient introduction and background were included with relevant references
Recommend adding hypotheses

Line 55 - Add references
Line 62 - Consider using variability consistently
Line 102 - What is going to be the highest value for the upper limit after analyzing with DFA? How was the FSI value of 1.30 determined?

Experimental design

Methods,

Recommend explaining about DFA procedures.

Line 124 - not a complete sentence, maybe a comma instead of a period
Line 156 - consider moving to the participants section
Line 201 - Please add a little bit more about the dependent measures means as the FSI stimuli chosen, how many FSI stimuli?

Validity of the findings

Results,

Exploring the upper limits of FSI in gait analysis is meaningfully supported by robust data, statistically sound.

How do you use gait variability data or any results related to gait variability?

Line 217 - Instead of commenting that the results are presented, it is necessary to describe the trend of the results with numbers, and then the table should be presented
Line 219 - It seems it is necessary to add more about the entrainment error in the introduction.

Discussion,

Line 236 - a little bit confusing, is it the time series of stride? Is it the time series of stride time? Or kinematic time series?
Line 260 - More knowledge-based explanations can be added to the introduction for better justification, This sentence shows the specific limit of 1.20

Conclusion,

Suggest making it more concise, focusing on the major findings,

Line 325 - This sentence can be moved to the Discussion
Line 332 - Recommend to move this sentence to Discussion as well

Additional comments

Overall, this manuscript addresses an interesting topic in gait variability. The research question is both relevant and well-defined. The study appears to be methodologically sound, with appropriate statistical analyses and clear presentation of results.

Annotated reviews are not available for download in order to protect the identity of reviewers who chose to remain anonymous.

·

Basic reporting

Basic Reporting
Introduction
1. Lines 44-48 reference specific metrics and findings throughout the literature but lack citations to support these notions. While the content is correct, if you say literature suggests something, it is critical to provide a few citations for this notion. For example, when referencing trends in the literature on the “noise within biological signals,” a few citations should back up this claim before transitioning into the specifics.
2. Great job setting the scene for the importance of DFA
3. I suggest splitting the sentence that runs from lines 61-65 into two sentences. It took me two reads to connect the ideas.
4. Line 91 – Can you add citations to support this thought? I’d consider
Schmid, D. G. (2024). Prospects of cognitive-motor entrainment: an interdisciplinary review. Frontiers in Cognition, 3, 1354116.
Thaut, M. H., McIntosh, G. C., & Hoemberg, V. (2015). Neurobiological foundations of neurologic music therapy: rhythmic entrainment and the motor system. Frontiers in psychology, 5, 1185.
Scott, N. M., Schmid, D., & Tomporowski, P. D. (2025). Effects of word presentation during treadmill walking on episodic memory and gait. Psychology of Sport and Exercise, 76, 102728.

I was very impressed with the author's writing in other sections, especially in the discussion.

Experimental design

Experimental Findings
Results
1. Please report PWS averages in the demographic information. This would be a nice cross-reference to the standard PWS for your participants (relative to their age) as a check to your protocol.
2. Why didn’t you test two more people to meet your sample size calculation after you identified that two participants' data would be excluded?
3. Conclusions are well stated, show the novelty of this work, and emphasize how this research can be used in the future.

Validity of the findings

Validity of Design
Methods: Participants Section
1. I am skeptical of the sample size calculation reported under the participants section (lines 109-111). Could you please report the software used and the parameters used to provide the results reported? Also, include the estimated effect size reported and a citation as to why this was chosen.
2. I’d suggest including exclusion criteria related to medications. Many medications can influence motor production and should be screened as an intervening variable. Additionally, inquiring about substance use and sleep patterns before participating in an experiment would be a nice addition to increase the validity of your findings.
3. Very clear FSI index analysis explanation. Well done.
Methods: Statistical analysis
1. I don’t love the use of 1.5 IQR as a statistical indicator of outliers. Because your sample size is so small, this isn’t the best measure to use. Additionally, a 1.5 IQR is quite liberal. I’d suggest using a more conservative measure of outliers and possibly pursuing a more rigorous technique such as mean absolute deviation (paper: Detecting outliers: Do not use standard deviation around the mean, use absolute deviation around the median - ScienceDirect)
Protocol
1. Ceiling limits to a PWS protocol are nice to have. In my lab, I’ve seen individuals report PWSs of over 4 mph. Moving forward, including a PWS cap at 3 mph (or whatever makes sense for your findings) would be a nice addition to the protocol used here.
2. I appreciate the attention to detail in the MATLAB auditory stimuli creation. Were participants listening to these signals through speakers or over headphones? Speakers were mentioned, but this was not clearly confirmed. (I’d be cautious of using Bluetooth headphones because of the increased information transmission time.) I’m just looking for a bit of clarity here for researchers who may want to use a similar research protocol.
3. It would be nice to see participant reports of RPE and HR (or another measure of cardiovascular output) during the PWS and research protocol to ensure that these physiological and psychological measures align with expectations. This would increase the reader’s confidence in the study’s findings.

Additional comments

I think this paper adds quite a bit of good information to future researchers seeking to use entrained gait protocols, as it exemplifies new techniques that could be used to effectively analyze gait output. However, I have strong concerns with the sample size selected and therefore slightly question the results. Overall, the paper does a nice job of setting the scene for the research, explaining methods and results, and discussing the implications of the findings. I greatly appreciate the different points the discussion section addresses while maintaining a concise and effective narrative. I would be curious to see if these results carry over to other age ranges and clinical populations! I’d encourage the authors to consider how cognition and motor performance go hand in hand and to explore some of this literature.

Reviewer 3 ·

Basic reporting

The English used in the manuscript is correct and precise. The article is included in the aim and scope of the PeerJ journal.
In the introduction, the authors cite very few studies and there is a lack of recent reports. The references presented are from 2020 and earlier, so there have been publications in the last 5 years in the field discussed by the authors, where auditory stimulation is used for gait stimulation. The introduction should be improved, new scientific reports from the field added, and a better description of the FSI problem.

Experimental design

The article is included in the aim and scope of the PeerJ journal.
The aim of the study is clearly stated. Nevertheless, by referring to literature from so many years ago, the innovation of the presented publication is questionable.
How much was the speed increased/decreased? Individual adaptation is one thing, but what was the jump in change? By 0.5 km/h or by 10% relative to PWS? Please describe this change methodology.
Why should patients synchronise their right foot movement? Left-sidedness should also be considered here by performing a simple test. Instructing a left-legged person to synchronise the step to the right leg causes dissonance and a forced movement. And in the study, the authors should have cared about natural movement (relatively speaking, as the treadmill also causes some forcing).
The quality of the sound generated from the loudspeaker is questionable. A better solution would have been to inflict the stimuli through headphones. Did the speaker use a connection to the computer via Bluetooth, or did the cable prepare everything?
If via Bluetooth, the delays generated by the transmission should also be considered. Then the synchronisation signal and the timing of the foot on the ground will naturally be shifted relative to each other, which can significantly distort the results.

Validity of the findings

How was the statistical analysis performed for the kinematic data? Was a comparison made between groups (for multiple groups)?
The discussion, like the theoretical introduction, should be enriched with the most recent scientific reports, as citing data from 10 years ago raises concerns about the quality of the work done and the reliability of the work to fill a scientific gap.

---

## Round 0.2 · accepted · Accept

The authors have addressed all reviewers' comments. The article is ready for publication.

Reviewer 1 ·

Basic reporting

Thank you for your effort to revise the manuscript following the reviewers' feedback. The authors have done a great job of incorporating the requested changes. The revised manuscript is more straightforward and easier to follow. The introduction, along with its references, provides a more compelling background. The methods and results section is better organized. The expanded discussion section has strengthened the paper's conclusion.

Experimental design

The experimental design is clearly defined, with revisions that provide relevant and sufficient information on study protocols.

Validity of the findings

Conclusions are made based on statistical analysis results and appropriately stated. .

Reviewer 3 ·

Basic reporting

The authors, as suggested, added the latest reports in the area of the represented topic, clarified the required concepts, and satisfactorily referred to the other comments.

Experimental design

No objections

Validity of the findings

No objections

Additional comments

No objections. Recommended for publication.